# Shoulder Pain and Trunk Muscles Endurance in Young Male and Female Swimmers

**DOI:** 10.3390/healthcare11152145

**Published:** 2023-07-27

**Authors:** Nikolaos Paramanidis, Athanasios Kabasakalis, Nikolaos Koutlianos, George Tsalis, Evangelia Kouidi

**Affiliations:** 1Laboratory of Sports Medicine, School of Physical Education and Sports Science at Thessaloniki, Aristotle University of Thessaloniki, 54124 Thessaloniki, Greece; nikosparamanidis@gmail.com (N.P.); koutlian@phed.auth.gr (N.K.); kouidi@phed.auth.gr (E.K.); 2Laboratory of Evaluation of Human Biological Performance, School of Physical Education and Sport Science at Thessaloniki, Aristotle University of Thessaloniki, 54124 Thessaloniki, Greece; 3School of Physical Education and Sports Science at Serres, Aristotle University of Thessaloniki, 54124 Thessaloniki, Greece; geotsa@gmail.com

**Keywords:** swimming, shoulder functional disability, swimmers’ shoulder, trunk muscles, young athletes

## Abstract

Shoulder pain is a common syndrome in swimming and affects a large number of competitive swimmers. The purpose of the study was to investigate the relationship between pain in the shoulder girdle and the endurance of the trunk muscles in young swimmers. A total of 24 boys and 22 girls, aged 13 to 18 years, participated in the study. The measurements included the completion of a questionnaire (Shoulder Pain and Disability Index, SPADI) and a field test (McGill’s Torso Muscular Endurance Test). The total SPADI score correlated weakly and negatively with the endurance time of back muscles in both sexes (r^2^ = 0.10, *p* = 0.035), and moderately and negatively in girls (r^2^ = 0.23, *p* = 0.023). A weak negative correlation was found between the disability index and the back muscles’ endurance time in both sexes (r^2^ = 0.15, *p* = 0.007), which was moderate in girls only (r^2^ = 0.25, *p* = 0.019). The disability index displayed moderate negative correlations with the right oblique’s (r^2^ = 0.18, *p* = 0.049) and left oblique’s endurance time (r^2^ = 0.23, *p* = 0.024) in girls. Weight, body mass index, the total out-of-water training time per week and age significantly affected the endurance times of the trunk muscles in boys and girls (*p* < 0.05). In conclusion, strengthening the dorsal and the oblique muscles could reduce shoulder pain and disability in young swimmers and especially girls.

## 1. Introduction

Swimming is a unique sport that combines endurance, strength, and physical control in an environment in which the use of body weight is drastically reduced [1]. Participation in competitive swimming is constantly increasing in terms of the number of swimmers [2]. Competitive-level swimmers may have five to seven training sessions per week and often double sessions in a day [3]. High-level swimmers can swim up to 14,000 m in one training session, which requires more than 2500 shoulder rotations per day or up to 16,000 rotations per week. The continuous use and overload of the upper limbs, with combined movements that pass above head level, provide most propulsive forces in all four known swimming techniques: the freestyle, breaststroke, backstroke, and butterfly. The percentage of upper limb use (relative to the lower limbs) in swimming can be as high as 90%, which could explain the cause of both the pain and injuries experienced by athletes. This huge rotational repetitiveness can easily lead to soft tissue and structural hyperextension syndromes in the shoulder area. This can result in severe pain during the athlete’s daily life and training sessions, even during rest [1]. This is also shown by 52% of high-level athletes reporting pain in the shoulder girdle, while the percentage of painful cases in athletes of a lower competitive level is estimated to be around 27%. This marked difference may be attributed to both an increased number of training hours and the long-term involvement in the sport that distinguishes high-level swimmers [2]. The incidence rates of so-called “swimmers’ shoulder” show that it can affect 91% of competitive swimmers [1]. Shoulder pain can become so severe that it can lead athletes to functional imbalances and eventually to the discontinuation of training. The conditions thought to cause pain in athletes include glenohumeral instability, rotator cuff muscle and bicep brachii muscle strains, and finally rotator cuff tendon impingement syndrome [3].

A lack of stability in the trunk muscles appears to play an active role in investigating both performance and injury in many athletes. It has been hypothesized that incomplete or poor core stability increases the likelihood of upper extremity injuries in professional athletes [4]. The core includes the muscles of the trunk and pelvis, which are responsible for maintaining the stability of the spine and the pelvic floor muscles. It also helps to generate and transfer energy from large to small parts of the body during athletic activities. Thus, the local stabilizing function of the core, as well as its ability to generate energy, is involved in all limb sports skills, such as running, throwing, kicking, and swimming [5]. Increasing core stability could be particularly beneficial for sprint swimmers and swimmers of different techniques, enabling the more efficient transfer of force from the trunk muscles to those of the upper and lower limbs, with the aim of propelling the body faster through the water [6].

There is a paucity of literature on the relationship between the strength and endurance of the trunk muscles and shoulder girdle function. This paper attempted to fill a gap in the relevant literature with regard to whether core muscle weakness negatively affects shoulder girdle function and whether low levels of trunk muscle endurance predispose someone to the occurrence of pain in the shoulder region. Thus, the purpose of this study was to investigate the relationship between shoulder girdle pain and shoulder functional disability, and the endurance of the trunk muscles in young swimmers. The subobjectives of the study were to examine the relationship between trunk muscle endurance, including the rectus abdominis, the lateral abdominal muscles bilaterally and the dorsal muscles, and reported shoulder zone pain, and make a comparative evaluation of the results in terms of sex.

## 2. Materials and Methods

### 2.1. Participants

A total of 46 highly trained swimmers (tier 3, according to the classification of MacKay et al. (2022) [7]), 24 boys and 22 girls aged 13 to 18 years (mean ± standard deviation, 14.7 ± 1.3 years), participated in the study. Twenty-three of the swimmers had freestyle as their preferred stroke, ten of them had breaststroke, six had backstroke, five had butterfly and two had individual medley. Sixteen swimmers had the 50 m as their preferred competitive distance, thirteen of them had the 100 m, eleven had the 200 m, one had the 400 m, two had the 800 m and three had the 1500 m. According to the classification of Ruiz-Navarro et al. (2023) [8], based on their performance, 7 swimmers were classified as level 3, 22 as level 4 and 17 as level 5. The participants were members of three competitive swimming clubs. Athletes eligible for enrolment were those aged between 13 and 18 years old, those not in the acute phase of any musculoskeletal pathology (no injuries present in the 3 weeks prior to the study) and those who did not participate in any form of training or warm-up prior to the measurement procedure. Their parents provided written informed consent that included all measurements’ information. The study was conducted according to the guidelines of the Declaration of Helsinki and approved by the Ethics Committee of the Department of Physical Education and Sport Science at Serres, Aristotle University of Thessaloniki (ERC-008/2023). The exclusion criteria were pain in the shoulder girdle in the acute phase, athletes receiving medication for musculoskeletal conditions and athletes out of the age range 13–18 years. 

### 2.2. Measurements

All procedures were carried out at each clubs’ training facilities. The measurements consisted of two parts: the completion of a questionnaire pertaining to shoulder girdle pain and shoulder functional disability, and the evaluation of the trunk muscles’ endurance using field tests. Prior to the questionnaire procedure, a member of the research group recorded the demographic and training characteristics of each participant (first and last name, sex, weight, height, age, predominant arm, training frequency/week in water, training frequency/week on land, training hours/session in water, training hours/session out of the water, main event and personal records). The questionnaire used was the Shoulder Pain and Disability Index (SPADI) [2], validated for Greek data, referring to the previous occurrence of shoulder pain. The athletes individually answered the questions of the questionnaire, which were divided into two categories, one indicative of pain (five questions with answers from 0 (no pain) to 10 (worst pain imaginable)), and the other indicative of functional disability (eight questions with answers from 0 (no difficulty) to 10 (so difficult to require help)). Responses were scaled from 1–10 in both subcategories of the questionnaire. To record the results of each athlete, the scores of each question were added together and then divided by the total number of questions multiplied by ten. At the end of the analysis, a percentage for each category of the questionnaire was calculated for each athlete. Percentages that exceeded 40% indicated significant pain or dysfunction in the shoulder girdle [9].

Following the questionnaire, the participants were assessed for trunk muscle endurance with the use of the McGill’s Torso Muscular Endurance Test [10]. According to the test, the swimmers performed a series of four isometric endurance exercises for the flexor muscles (rectus abdominis) or anterior abdominals, the lateral abdominal (oblique) muscles (one for each side of the body) and extensor (dorsal) muscles. Athletes were required to be fully rested before performing the tests, so they did not perform any warm-ups, stretching, or prior training on the day of measurements. Specifically, this research protocol took place 24 h after their last training session. The goal of each exercise was to allow the swimmers to maintain the appropriate isometric posture for as long as they could without deviating from it, which would lead to the completion of each exercise test. All participants were allowed to familiarize themselves with the procedure and they also started each test whenever they felt ready, with the examiner starting the timer from the moment they took the appropriate position [10]. The examiner provided verbal instructions to the swimmers to assist and motivate them in order to maintain their position. The time was measured by means of an electronic stopwatch (Finis, Livermore, CA, USA) and all the tests were performed on a therapeutic bed. The rest time after each exercise was two minutes to allow for adequate recovery. 

Regarding the tests, in the first one, during which the anterior abdominal (flexor) muscles’ endurance was tested, participants sat on the bed in a long sitting position, while their legs were secured with straps at the level of the ankles. Knees and hips were brought to an angle of 90° and the arms were hugging the shoulders with the elbows in an imaginary line with the chest. Then, with the aid of an analogue goniometer, the examiner brought the participant’s torso to 60°, with the subject trying to maintain this position for as long as possible. The test was terminated when the participant was unable to maintain the 60° position and moved their torso more than 10 cm from the appropriate position in the oblique plane [10].

In the second and third tests, during which the endurance of the lateral abdominal muscles was tested, participants were placed in a lateral plank position on the left and right arm, respectively, in a random order. The elbow and forearm resting on the bed had to be in an imaginary straight line with the respective shoulder, while the rest of the body had to be in a straight line with the pelvis raised from the bed. Participants maintained this position until the pelvis began to show obvious signs of change and increased movement in the frontal and transverse planes, which terminated the test [10]. 

Finally, to test the dorsal (extensor) muscles’ endurance, participants were placed in a prone position on the bed, bringing their pelvis to the exact corner of the bed, with their legs extended and their torso and arms touching the floor. At the start of the test, the participants performed trunk extension with their arms hugging their shoulders until the trunk came into line with the bed; the participants’ legs were restrained by the examiner, who observed any significant changes in the trunk from the correct position, at which point they would end the test [10]. 

### 2.3. Statistical Analysis

G*Power analysis was performed before the study by setting an effect size of 0.75, a probability error of 0.05, and a power of 0.8 for two groups. The analysis indicated that 46 subjects was the smallest acceptable number of participants. The distribution of the data was examined using the Shapiro–Wilk test. For the comparison between boys and girls, we used an independent samples *t*-test when there was a normal distribution of data and a Mann–Whitney U test when the normality of data was not met. Linear regression models per two variables were used for correlation analysis. Statistical models were applied using R v4.0.3 in Rstudio v.1.3.1093, where chart design was performed. In each model, the r^2^ and *p*-value were indicated to explore the strength of correlation and the statistical significance of the model, respectively. Correlation was interpreted as negligible (r < 0.10), weak (r—0.10–0.39), moderate (r = 0.40–0.69), strong (r = 0.70–0.90) and very strong (r—0.91–1.00) [11]. Means and standard deviations were calculated for all key variables. We set the level of statistical significance at α = 0.05.

## 3. Results

Table 1 includes the demographic and training characteristics of the participants. Moreover, the results of the SPADI questionnaire and the trunk muscle endurance tests are presented in Table 2. 

Linear regression provided few significant negative correlations between the main parameters analyzed (SPADI questionnaire and McGill’s test outcomes), as shown in Table 3.

Additional correlations were also found between the demographic and training characteristics of the swimmers, and the SPADI questionnaire and McGill test outcomes in the whole group of boys and girls. Body weight correlated weakly and negatively with the flexor and extensor muscles’ endurance time (r^2^ = 0.11, *p* = 0.028 and r^2^ = 0.13, *p* = 0.015, respectively). BMI correlated moderately negatively and out of the water training time correlated weakly positively with the flexor muscles’ endurance time (r^2^ = 0.18, *p* = 0.004 and r^2^ = 0.09, *p* = 0.049, respectively). BMI also correlated moderately negatively with the extensor muscles’ endurance time (r^2^ = 0.16, *p* = 0.006) and out of the water training time correlated weakly negatively with the disability score (r^2^ = 0.09, *p* = 0.46). 

## 4. Discussion

In the present study, we examined the possible relationship between the pain and disability of previous shoulder incidents, as expressed in the SPADI questionnaire, and the endurance of the trunk muscles, as assessed using the McGill test, in adolescent competitive swimmers of both sexes. Although no significant correlations were found between shoulder pain and the trunk muscles’ endurance times, the functional disability score was negatively correlated with the dorsal (extensors) muscles’ endurance. The disability score was also negatively correlated with the lateral abdominal muscles’ (right and left obliques) endurance in girls. Moreover, variables such as weight, BMI, and total out-of-water training time per week were found to be correlated with the endurance of the flexor or extensor trunk muscles of the swimmers, indicating that the trunk muscles’ endurance may be affected by the anthropometric or training characteristics of the swimmers.

The absence of any correlation between the pain score and trunk muscles’ endurance capacity possibly questions the notion that the trunk muscles’ endurance affects the existence and severity of shoulder pain. However, the functional disability score correlated with the extensor trunk muscles’ endurance, mainly in girls. Moreover, the lateral abdominal muscles’ endurance was correlated with the disability score in girls. These findings possibly indicate that the relationship between the trunk muscles’ endurance and shoulder disability is sex-specific. Thus, at least in girls, the existence of shoulder disability may be influenced by the dorsal or lateral flexor muscles’ endurance. The strengthening of the dorsal muscles seems to prevent the occurrence of shoulder injuries, as shown by several studies in the literature [1,3,5,12]. Moreover, since it has been recently found that, in overhead athletes with shoulder pain, the lateral flexor muscles affect the functional upper limb performance [13], the strengthening of these muscles seems also worthy of consideration.

In general, athletes need to properly align their body and functionally connect the core to the upper and lower extremities. It also appears in adolescent athletes that the muscles of the posterior chain are often less developed than those of the anterior chain. This results in the reduced control of their body during basic functional movements and physical activity [12,14]. This fact is possibly related to the increased functional disability of the shoulder in adolescent female swimmers when the dorsal muscles’ endurance capacity was lower than the flexors’, as was found in the present study. Tate et al. [3] studied 236 female high-level swimmers aged 12 to 14 years. In their study, girls with pain and functional disability in the shoulder girdle showed significantly reduced trunk muscle endurance (measured using a hand-held chronometer during lateral bridges) than their less symptomatic co-athletes, indicating that pain in the shoulder girdle seems to co-exist with reduced endurance in the trunk muscles. However, it is not clear whether lower levels of endurance in the trunk muscles are the cause of this phenomenon. A possible confounding factor is the increased predisposition of girls to pain in the shoulder girdle, as reported by Harrington et al. [15].

Considering swimming performance in particular, Weston et al. [6] demonstrated an improvement in the core endurance time during the ‘bridge’ exercise, an increase in maximal electro-myographic activity in the core muscles via strengthening exercises mainly directed at the core, and an increase in the shoulder girdle (prone bridge, side bridge, bird dog, leg raise, overhead squat, sit twist, shoulder press) in pre-adolescent and adolescent athletes. Apart from pointing out the relationship between core muscles and shoulder girdle muscles in swimmers, this resulted in better individual swimming times in the 50 m freestyle by 2% compared to the times of the athletes in the control group. Furthermore, Matsuuda et al. (2023) [16] highlighted the need for improvements in the coordination of the upper limbs and the trunk muscles in swimmers with “swimmers’ shoulder”. This might indicate the connection between the trunk muscles and shoulder joint muscles, which in turn possibly explains the relationship between the trunk muscles’ endurance and shoulder disability observed in the present study.

Regarding the secondary analyses of the study, no significant correlation between the key parameters and the total weekly training time in water was found in both sexes. This finding suggests that the presence of shoulder pain and shoulder functional disability, as well as the trunk muscles’ endurance, were not related to the total weekly swimming training time in the athletes who participated in the study. This finding is equivocal to previous findings in the literature that attribute the cause of the existence of pain to the high repetitiveness of shoulder girdle movements, resulting in the degeneration of soft tissues [1] and the tendons of the rotator cuff muscles as they impinge and come into continuous contact with bony prominences such as the acromion. In a recent study, shoulder injuries were linked to total training distances in competitive swimmers [17]. Perhaps, the volume of swimming training that the swimmers of this study undertook did not exceed a certain limit that could cause shoulder pain. Alternatively, it could be argued that the progression of training to that volume contributed to the absence of shoulder pain or disability, as has been found elsewhere [18].

The total out-of-water training correlated significantly with the abdominal flexor trunk muscles’ endurance in both boys and girls. This likely shows that conditioning outside of the pool positively affects the trunk muscles’ endurance, but only in part, as no other correlations were found. A greater focus on the strengthening of the flexor trunk muscles during out-of-water training in this group of swimmers is possibly the cause of this finding, suggesting that the training program should be readjusted to stress the other trunk muscles as well. This is especially considering that it was the extensors’ and lateral flexor muscles’ endurance times that correlated with the disability score, at least in girls. However, a correlation between the functional disability score and total on-land training time per week was found in the whole group of swimmers. Thus, as the time spent training on land using resistance exercises or circuit training increased, the shoulder disability reported by the young athletes decreased. This finding suggests a likely positive effect of on-land training in adolescent swimmers regarding shoulder disability. Out-of-water training has been also found recently to affect the prevalence of swimmer’s shoulder [19]. In addition, the fact that girls in the present study spent less time on land training may explain in part the pronounced correlations found in girls between shoulder functional disability and the trunk muscles’ endurance times. 

As the habits of adolescent club swimmers have been found to affect shoulder pain and disability [20], an appropriate focus on suitable out-of-water training would potentially aid in the prevention of shoulder incidents. Since the out-of-water training performed by the participants in our study mainly included a variety of preventive, functional, resistance and/or circuit training, using body weight, elastic bands, suspension training and/or free weights, according to their reports, a handful of shoulder pain and disability “preventive habits” could be included in the programs of young swimmers. Some additional correlations found were the negative ones between weight and BMI and both flexor and extensor muscles’ endurance times, possibly displaying the importance of the impact of the anthropometric characteristics of the swimmers on the conditioning of the trunk muscles and the prescription of out-of-water training for the conditioning of trunk muscles.

The outcomes of the present study should be interpreted in light of the age of the participants. The trunk muscles’ endurance is positively associated with biological maturity from 13 years onwards, and during adolescence, the trunk muscles’ endurance has been found to reach a peak, with differences between boys and girls, mostly in favor of the boys [21,22]. Adolescents respond to sports training with strength gains and muscle hypertrophy, and the strength of trunk flexors and extensors has been found to correlate with the cross-sectional area of the muscle, adjusted for body mass [23]. Moreover, trunk muscle training has been found to be effective in improving specific measures and performance in young athletes [24]. Such outcomes have previously been confirmed in girls [25]. Recently, in a study involving adolescent athletes (and swimmers among them), it was found that the trunk muscles’ strength is affected by age, sex and sport specialization [26]. Based on these available data, we suggest that the findings and applications of the study are considered from the perspective of the sensitive period of adolescence.

Our study has some potential limitations, such as the subjective nature of the questionnaires and the swimmers’ perceptions regarding pain or disability. Additionally, the study lacked more detailed measurements of the trunk muscles’ endurance using tools that could better record the athletes’ effort, such as an EMG feedback device. However, we wished to employ tools and tests that could be easily used by practitioners at the training facilities of the athletes.

## 5. Practical Applications

The strengthening of the dorsal muscles in young female swimmers may improve the functional capacity of the shoulder girdle, as it appears to be a tool for preventing shoulder disability.Adolescent male and female swimmers may differ in terms of the association between shoulder pain and disability, and the trunk muscles’ endurance.Out-of-water training may be usefully employed in order to improve the conditioning of trunk muscles and subsequently reduce the levels of functional impairment in the shoulder girdle of young swimmers.

## 6. Conclusions

In summary, the results of the present study demonstrated that the shoulder functional disability score, as measured using the SPADI questionnaire, is affected by the dorsal extensor trunk muscles’ endurance, as assessed using the McGill test, mainly in female adolescent swimmers. Therefore, young female swimmers and their coaches are encouraged to include more relevant exercises in order to improve the dorsal trunk muscles’ endurance. Moreover, our results indicated that the out-of-water training time was positively correlated with the abdominal flexor trunk muscles’ endurance, but not with the dorsal trunk muscles’ endurance. Of course, the causes of dorsal trunk muscle endurance deficits in girls with a functional disability in the shoulder girdle remain unclear and further studies are required in order to fully investigate the mechanisms that associate trunk muscle endurance with shoulder disability. Nevertheless, it can be fairly argued that conditioning the muscles of the posterior motor chain may reduce the chances of experiencing shoulder disability, especially in young female swimmers. 

## Figures and Tables

**Table 1 healthcare-11-02145-t001:** Demographic and training characteristics of the swimmers (mean and standard deviation, SD).

	Boys		Girls	
	Mean	SD	Mean	SD
Age (years)	14.8	0.26	14.5	0.26
Weight (kg)	64.1 *	1.64	52.9	1.38
Height (cm)	175.1 *	1.71	163.7	1.22
BMI (kg/m^2^)	20.8	3.27	19.7	3.98
*Weekly training*				
Swimming (min)	654	28	645	18
Out of water (min)	211 *	16	165	12
Total (min)	865	38	810	20
Competitive level (FINA points)	470	26	522	24

BMI: Body mass index. * Significant difference vs. girls (*p* < 0.05).

**Table 2 healthcare-11-02145-t002:** Results of SPADI questionnaire responses and trunk muscle endurance times (McGill’s Test) of the swimmers (mean and SD).

	Boys (n = 24)		Girls (n = 22)	
	Mean	SD	Mean	SD
Total pain Score (%)	22	4	25	4
Total disability Score (%)	6	2	10	2
Total SPADI Score (%)	12	2	16	3
Flexors’ time (s)	86.16 *	6.05	122.95	15.37
Extensors’ time (s)	94.92	6.82	104.27	8.24
Right side oblique’s time (s)	87.56	5.98	91.63	7.26
Left side oblique’s time (s)	78.76	5.34	79.81	5.72

Flexors’ time: anterior abdominals’ endurance time; Extensors’ time: dorsal muscles endurance time. * Significant difference vs. girls (*p* < 0.05).

**Table 3 healthcare-11-02145-t003:** Linear regression analysis results (r^2^ and *p*) between SPADI questionnaire and McGill’s test outcomes).

	Pain Score			Disability Score			Total SPADI Score		
	Boys	Girls	All	Boys	Girls	All	Boys	Girls	All
Flexors’ time	ns	ns	ns	ns	ns	ns	ns	ns	ns
Extensors’ time	ns	ns	ns	ns	r^2^ 0.25 *p* 0.019 **	r^2^ 0.15 *p* 0.007 *	ns	r^2^ 0.23 *p* 0.023 **	r^2^ 0.10 *p* 0.035 *
Right oblique’s time	ns	ns	ns	ns	r^2^ 0.18 *p* 0.049 **	ns	ns	ns	ns
Left oblique’s time	ns	ns	ns	ns	r^2^ 0.23 *p* 0.024 **	ns	ns	ns	ns

Flexors’ time: anterior abdominals’ endurance time; Extensors’ time: dorsal muscles’ endurance time.; ns: not significant. * Weak correlation; ** moderate correlation [11].

## Data Availability

Data is unavailable due to privacy or ethical restrictions.

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
