# Peer review of "Shoulder Pain and Trunk Muscles Endurance in Young Male and Female Swimmers"

_healthcare, 2023, doi:10.3390/healthcare11152145_

Round 1

Reviewer 1 Report

Thank you for the opportunity to review this study assessing the relationship between shoulder pain and trunk muscle endurance in adolescent swimmers. Following are edits to improve the presentation of the study.

General: change gender to sex throughout since you have classified boys and girls based on biology. Along these same lines, use boys instead of males and girls instead of females.

Abstract

Reword the first sentence for clarification: "Shoulder pain is a common orthopaedic symptom in competitive swimmers."

Report results (r values, actual p values) for the SPADI/extensors, SPADI/flexors, SPADI/LOblique, and SPADI/ROblique relationships as word count allows; state the strength and direction of the significant associations

The conclusion that "strength training may be linked with limited possibility of...."is awkwardly worded. Be specific with what group(s) of muscles, if strengthened, could reduce pain and disability, especially in girls.

Introduction

Line 30-31: This statement is unclear; please reword: Participation in 30 competitive swimming is constantly increasing, with a peak in age when major events take place

Solid points made in the Introduction that logically lead to the purpose; research is well cited.

Materials and Methods

Please define what moderately and highly trained mean as this can significantly affect you results. How many years competitive experience? What were the breakdown by stroke dominance (I understand at this age they may not be specialized yet; at least can you provide some description as to sprint, middle distance, distance?)?

This section has good detail on how testing was conducted. However, please clarify the type of "electronic stopwatch" used in line 120, and consider including photos of the trunk muscle endurance tests.

Were there any issues with youth and maximal effort? What verbal instructions did you provide the subjects in performing the tests so as to ensure comprehension and adequate effort?

In section 2.3, state how the correlation strength will be assessed and cite a reference for this

Results

Please state the strength of the significant correlations (e.g., weak).

Discussion

Key findings are discussed in relationship to previous research - much of which is on adult swimmers though; I suggest including some commentary on growth and maturation regarding trunk muscle strength and how that may have impacted the results.

References

Please update the "in press" for references 11 and 12 as the works are now published.

Author Response

Response to comments:

Reviewer 1

Comment: Thank you for the opportunity to review this study assessing the relationship between shoulder pain and trunk muscle endurance in adolescent swimmers. Following are edits to improve the presentation of the study.

Response: We thank you for your useful comments.

General

Comment: change gender to sex throughout since you have classified boys and girls based on biology. Along these same lines, use boys instead of males and girls instead of females.

Response: We replaced gender with sex and used boys and girls instead of males and females (where appropriate) throughout the manuscript.

Abstract

Comment: Reword the first sentence for clarification: "Shoulder pain is a common orthopaedic symptom in competitive swimmers."

Response: We reworded this sentence (l. 13).

Comment: Report results (r values, actual p values) for the SPADI/extensors, SPADI/flexors, SPADI/LOblique, and SPADI/ROblique relationships as word count allows; state the strength and direction of the significant associations.

Response: We reported r2 and p values for all the significant relationships between SPADI questionnaire and McGill’s test parameters, as well as the strength and direction of them (l. 18-23). However, this led the word count to exceed the 200 words limit.

Comment: The conclusion that "strength training may be linked with limited possibility of...."is awkwardly worded. Be specific with what group(s) of muscles, if strengthened, could reduce pain and disability, especially in girls.

Response: We rephrased the conclusion accordingly (l. 25-26).

Introduction

Comment: Line 30-31: This statement is unclear; please reword: Participation in competitive swimming is constantly increasing, with a peak in age when major events take place.

Response: We reworded this sentence (l. 31-32).

Comment: Solid points made in the Introduction that logically lead to the purpose; research is well cited.

Response: Thank you for the comment.

Materials and Methods

Comment: Please define what moderately and highly trained mean as this can significantly affect you results. How many years competitive experience? What were the breakdown by stroke dominance (I understand at this age they may not be specialized yet; at least can you provide some description as to sprint, middle distance, distance?)?

Response: We specified further the training and performance status of the participants according to the literature and available data (l. 77-85).

Comment: This section has good detail on how testing was conducted. However, please clarify the type of "electronic stopwatch" used in line 120, and consider including photos of the trunk muscle endurance tests.

Response: We added the type of the electronic stopwatch used (l. 130). Lacking rights to publish photos prevented us from including any photos in our paper. However, the tests used in our study are well presented in the referenced source (10).

Comment: Were there any issues with youth and maximal effort? What verbal instructions did you provide the subjects in performing the tests so as to ensure comprehension and adequate effort?

Response: Since the participants were competitive athletes, familiar with maximal efforts, there were no issues with them performing the tests. The verbal instructions provided, included instructions of how to perform the exercises correctly along with some motivational words in order to ensure they reached their maximal times. The examiner tried to keep feedback and same motivation for all participants. We included a relevant sentence in the text (l. 128-129).

Comment: In section 2.3, state how the correlation strength will be assessed and cite a reference for this.

Response: We inserted the strength range of the correlation, based on a relevant reference (l. 166-168).

Results

Comment: Please state the strength of the significant correlations (e.g., weak).

Response: We stated the strength of the significant correlations (table 3 and l. 191-197).

Discussion

Comment: Key findings are discussed in relationship to previous research - much of which is on adult swimmers though; I suggest including some commentary on growth and maturation regarding trunk muscle strength and how that may have impacted the results.

Response: Thank you for your comment. We included a new relevant paragraph as suggested (l. 292-304).

References

Comment: Please update the "in press" for references 11 and 12 as the works are now published.

Response: These references (now 15 and 16) are still “online ahead of print”.

Reviewer 2 Report

First of all, congratulations for the research work carried out, I would like to highlight the originality of the research question. In addition, I am aware of how difficult it is to carry out research without funding support. Next, I will mention a series of recommendations in order to obtain clearer and more accurate information on your results.

1-Authors are advised to evaluate the power of the sample with the G-Power for example.

2-In the material and methods, it mentions that the T-test for independent samples will be used, but it does not clarify whether prior tests are carried out to confirm that you can use a parametric test such as Shapiro-Wilk or Kolmogorov-Smirnov. Indicate this in the statistical analysis section or otherwise apply non-parametric tests.

3-It is recommended to correlate the main swimming style of each swimmer with the rest of the variables.

4-It would be interesting to go deeper into the type of training performed outside the pool.

5-In the discussion section, I encourage the inclusion of more references that allow a more in-depth comparison and discussion of the data obtained.

I hope these observations will be helpful for your original study.

My best regards.

Author Response

Reviewer 2

Comment: First of all, congratulations for the research work carried out, I would like to highlight the originality of the research question. In addition, I am aware of how difficult it is to carry out research without funding support. Next, I will mention a series of recommendations in order to obtain clearer and more accurate information on your results.

Response: We thank you for your kind and useful comments.

Comment 1-Authors are advised to evaluate the power of the sample with the G-Power for example.

Response: Thank you for your comment. The G*Power analysis is now included in the manuscript (l. 157-159).

Comment 2-In the material and methods, it mentions that the T-test for independent samples will be used, but it does not clarify whether prior tests are carried out to confirm that you can use a parametric test such as Shapiro-Wilk or Kolmogorov-Smirnov. Indicate this in the statistical analysis section or otherwise apply non-parametric tests.

Response: We have now added the analysis of the distribution of our data with the Shapiro-Wilk test. Where normality was not met, we used the Mann-Whitney U test instead of the independent samples t test (l. 159-162).

Comment 3-It is recommended to correlate the main swimming style of each swimmer with the rest of the variables.

Response: This would indeed be very interesting in older and more experienced swimmers, who would have established a standard main stroke and/or event and would train more specifically for it. Responding to the other reviewer, we included the preferred strokes and competitive distances of the swimmers, without stating them as main strokes or events, but as preferred ones, as the training programs of the participants were not so specialized yet due to their age and status (see about their classification in l. 77-85). Moreover, the distribution of the preferred strokes of the participants would not allow us to reach to clear conclusions. Thus, we considered that it would be better not to include this analysis.

Comment 4-It would be interesting to go deeper into the type of training performed outside the pool.

Response: We added some available rough descriptive information (l. 283-287), without going much deeper however, as the out of the water training of the swimmers was not monitored thoroughly by the examiners.

Comment 5-In the discussion section, I encourage the inclusion of more references that allow a more in-depth comparison and discussion of the data obtained.

Response: We added more referenced text that expanded our discussion and hopefully assisted the interpretation of our findings (l.220-222, 276-277, 292-304).

Comment: I hope these observations will be helpful for your original study.

Response: Thank you. They were indeed helpful.